# Best of Both Worlds: Transferring Knowledge from Discriminative Learning to a Generative Visual Dialog Model

**Jiasen Lu**[1]*, **Anitha Kannan**[2]*, **Jianwei Yang**[1], **Devi Parikh**[3,1], **Dhruv Batra**[3,1]
[1] Georgia Institute of Technology, [2] Curai, [3] Facebook AI Research
{jiasenlu, jw2yang, parikh, dbatra}@gatech.edu

## Abstract

We present a novel training framework for neural sequence models, particularly for grounded dialog generation. The standard training paradigm for these models is maximum likelihood estimation (MLE), or minimizing the cross-entropy of the human responses. Across a variety of domains, a recurring problem with MLE trained generative neural dialog models ($G$) is that they tend to produce 'safe' and generic responses (*'I don't know'*, *'I can't tell'*). In contrast, discriminative dialog models ($D$) that are trained to rank a list of candidate human responses outperform their generative counterparts; in terms of automatic metrics, diversity, *and* informativeness of the responses. However, $D$ is not useful in practice since it can not be deployed to have real conversations with users.

Our work aims to achieve the best of both worlds – the practical usefulness of $G$ and the strong performance of $D$ – via knowledge transfer from $D$ to $G$. Our primary contribution is an end-to-end trainable generative visual dialog model, where $G$ receives gradients from $D$ as a *perceptual* (not adversarial) loss of the sequence sampled from $G$. We leverage the recently proposed Gumbel-Softmax (GS) approximation to the discrete distribution – specifically, a RNN augmented with a sequence of GS samplers, coupled with the straight-through gradient estimator to enable end-to-end differentiability. We also introduce a stronger encoder for visual dialog, and employ a self-attention mechanism for answer encoding along with a metric learning loss to aid $D$ in better capturing semantic similarities in answer responses. Overall, our proposed model outperforms state-of-the-art on the VisDial dataset by a significant margin (2.67% on recall@10).

## 1 Introduction

One fundamental goal of artificial intelligence (AI) is the development of perceptually-grounded dialog agents – specifically, agents that can perceive or understand their environment (through vision, audio, or other sensors), and communicate their understanding with humans or other agents in natural language. Over the last few years, neural sequence models (*e.g.* [47, 44, 46]) have emerged as the dominant paradigm across a variety of setting and datasets – from text-only dialog [44, 40, 23, 3] to more recently, visual dialog [7, 9, 8, 33, 45], where an agent must answer a sequence of questions grounded in an image, requiring it to reason about both visual content and the dialog history.

The standard training paradigm for neural dialog models is maximum likelihood estimation (MLE) or equivalently, minimizing the cross-entropy (under the model) of a 'ground-truth' human response. Across a variety of domains, a recurring problem with MLE trained neural dialog models is that they tend to produce 'safe' generic responses, such as *'Not sure'* or *'I don't know'* in text-only dialog [23], and *'I can't see'* or *'I can't tell'* in visual dialog [7, 8]. One reason for this emergent behavior is that

the space of possible next utterances in a dialog is *highly* multi-modal (there are many possible paths a dialog may take in the future). In the face of such highly multi-modal output distributions, models 'game' MLE by latching on to the head of the distribution or the frequent responses, which by nature tend to be generic and widely applicable. Such safe generic responses break the flow of a dialog and tend to disengage the human conversing with the agent, ultimately rendering the agent useless. It is clear that novel training paradigms are needed; that is the focus of this paper.

One promising alternative to MLE training proposed by recent work [36, 27] is *sequence-level training* of neural sequence models, specifically, using reinforcement learning to optimize task-specific sequence metrics such as BLEU [34], ROUGE [24], CIDEr [48]. Unfortunately, in the case of dialog, *all existing* automatic metrics correlate poorly with human judgment [26], which renders this alternative infeasible for dialog models.

In this paper, inspired by the success of adversarial training [16], we propose to train a *generative* visual dialog model ($G$) to produce sequences that score highly under a *discriminative* visual dialog model ($D$). A discriminative dialog model receives as input a candidate list of possible responses and learns to sort this list from the training dataset. The generative dialog model ($G$) aims to produce a sequence that $D$ will rank the highest in the list, as shown in Fig. 1.

Note that while our proposed approach is inspired by adversarial training, there are a number of subtle but crucial differences over generative adversarial networks (GANs). Unlike traditional GANs, one novelty in our setup is that our discriminator receives a list of candidate responses and explicitly learns to reason about similarities and differences across candidates. In this process, $D$ learns a task-dependent perceptual similarity [12, 19, 15] and learns to recognize multiple correct responses in the feature space. For example, as shown in Fig. 1 right, given the image, dialog history, and question *'Do you see any bird?'*, besides the ground-truth answer *'No, I do not'*, $D$ can also assign high scores to other options that are valid responses to the question, including the one generated by $G$: *'Not that I can see'*. The interaction between responses is captured via the similarity between the learned embeddings. This similarity gives an additional signal that $G$ can leverage in addition to the MLE loss. In that sense, our proposed approach may be viewed as an instance of 'knowledge transfer' [17, 5] from $D$ to $G$. We employ a metric-learning loss function and a self-attention answer encoding mechanism for $D$ that makes it particularly conducive to this knowledge transfer by encouraging perceptually meaningful similarities to emerge. This is especially fruitful since prior work has demonstrated that discriminative dialog models significantly outperform their generative counterparts, but are not as useful since they necessarily need a list of candidate responses to rank, which is only available in a dialog dataset, not in real conversations with a user. In that context, our work aims to achieve the best of both worlds – the practical usefulness of $G$ and the strong performance of $D$ – via this knowledge transfer.

Our primary technical contribution is an end-to-end trainable generative visual dialog model, where the generator receives gradients from the discriminator loss of the sequence sampled from $G$. Note that this is challenging because the output of $G$ is a sequence of discrete symbols, which naïvely is not amenable to gradient-based training. We propose to leverage the recently proposed Gumbel-Softmax (GS) approximation to the discrete distribution [18, 30] – specifically, a Recurrent Neural Network (RNN) augmented with a sequence of GS samplers, which when coupled with the straight-through gradient estimator [2, 18] enables end-to-end differentiability.

Our results show that our 'knowledge transfer' approach is indeed successful. Specifically, our discriminator-trained $G$ outperforms the MLE-trained $G$ by 1.7% on recall@5 on the VisDial dataset, essentially improving over state-of-the-art [7] by 2.43% recall@5 and 2.67% recall@10. Moreover, our generative model produces more diverse and informative responses (see Table 3).

As a side contribution specific to this application, we introduce a novel encoder for neural visual dialog models, which maintains two separate memory banks – one for visual memory (where do we look in the image?) and another for textual memory (what facts do we know from the dialog history?), and outperforms the encoders used in prior work.

## 2   Related Work

**GANs for sequence generation.** Generative Adversarial Networks (GANs) [16] have shown to be effective models for a wide range of applications involving continuous variables (*e.g.* images) *c.f* [10, 35, 22, 55]. More recently, they have also been used for discrete output spaces such as language generation – *e.g.* image captioning [6, 41], dialog generation [23], or text generation [53] – by either viewing the generative model as a stochastic parametrized policy that is updated using REINFORCE

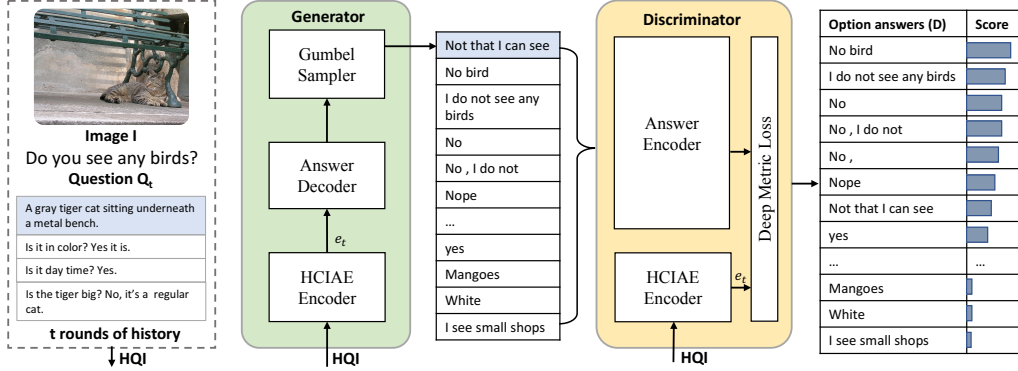

**Figure 1:** Model architecture of the proposed model. Given the image, history, and question, the discriminator receives as additional input a candidate list of possible responses and learns to sort this list. The generator aims to produce a sequence that discriminator will rank the highest in the list. The right most block is $D$'s score for different candidate answers. Note that the multiple plausible responses all score high. Image from the COCO dataset [25].

with the discriminator providing the reward [53, 6, 41, 23], or (closer to our approach) through continuous relaxation of discrete variables through Gumbel-Softmax to enable backpropagating the response from the discriminator [21, 41].

There are a few subtle but significant differences w.r.t. to our application, motivation, and approach. In these prior works, both the discriminator and the generator are trained in tandem, and from scratch. The goal of the discriminator in those settings has primarily been to discriminate 'fake' samples (*i.e.* generator's outputs) from 'real' samples (*i.e.* from training data). In contrast, we would like to transfer knowledge from the discriminator to the generator. We start with pre-trained $D$ and $G$ models suited for the task, and then transfer knowledge from $D$ to $G$ to further improve $G$, while keeping $D$ fixed. As we show in our experiments, this procedure results in $G$ producing diverse samples that are close in the embedding space to the ground truth, due to perceptual similarity learned in $D$. One can also draw connections between our work and Energy Based GAN (EBGAN) [54] – without the adversarial training aspect. The "energy" in our case is a deep metric-learning based scoring mechanism, instantiated in the visual dialog application.

**Modeling image and text attention.** Models for tasks at the intersection of vision and language – *e.g.*, image captioning [11, 13, 20, 49], visual question answering [1, 14, 31, 37], visual dialog [7, 9, 8, 45, 33] – typically involve attention mechanisms. For image captioning, this may be attending to relevant regions in the image [49, 51, 28]. For VQA, this may be attending to relevant image regions alone [4, 50, 52] or co-attending to image regions and question words/phrases [29].

In the context of visual dialog, [7] uses attention to identify utterances in the dialog history that may be useful for answering the current question. However, when modeling the image, the entire image embedding is used to obtain the answer. In contrast, our proposed encoder HCIAE (Section 4.1) localizes the region in the image that can help reliably answer the question. In particular, in addition to the history and the question guiding the image attention, our visual dialog encoder also reasons about the history when identifying relevant regions of the image. This allows the model to implicitly resolve co-references in the text and ground them back in the image.

## 3 Preliminaries: Visual Dialog

We begin by formally describing the visual dialog task setup as introduced by Das *et al.* [7]. The machine learning task is as follows. A visual dialog model is given as input an image $\boldsymbol{I}$, caption $\boldsymbol{c}$ describing the image, a dialog history till round $t-1$, $\boldsymbol{H} = (\underbrace{\boldsymbol{c}}_{H_0}, \underbrace{(\boldsymbol{q}_1, \boldsymbol{a}_1)}_{H_1}, \ldots, \underbrace{(\boldsymbol{q}_{t-1}, \boldsymbol{a}_{t-1})}_{H_{t-1}})$, and the followup question $\boldsymbol{q}_t$ at round $t$. The visual dialog agent needs to return a valid response to the question.

Given the problem setup, there are two broad classes of methods – generative and discriminative models. Generative models for visual dialog are trained by maximizing the log-likelihood of the ground truth answer sequence $\boldsymbol{a}_t^{gt} \in \mathcal{A}_t$ given the encoded representation of the input $(\boldsymbol{I}, \boldsymbol{H}, \boldsymbol{q}_t)$.

On the other hand, discriminative models receive both an encoding of the input $(\boldsymbol{I}, \boldsymbol{H}, \boldsymbol{q}_t)$ *and* as additional input a list of 100 candidate answers $\mathcal{A}_t = \{\boldsymbol{a}_t^{(1)}, \ldots, \boldsymbol{a}_t^{(100)}\}$. These models effectively learn to sort the list. Thus, by design, they cannot be used at test time without a list of candidates available.

# 4 Approach: Backprop Through Discriminative Losses for Generative Training

In this section, we describe our approach to transfer knowledge from a discriminative visual dialog model ($D$) to generative visual dialog model ($G$). Fig. 1 (a) shows the overview of our approach. Given the input image $\boldsymbol{I}$, dialog history $\boldsymbol{H}$, and question $\boldsymbol{q}_t$, the encoder converts the inputs into a joint representation $\boldsymbol{e}_t$. The generator $G$ takes $\boldsymbol{e}_t$ as input, and produces a distribution over answer sequences via a recurrent neural network (specifically an LSTM). At each word in the answer sequence, we use a Gumbel-Softmax sampler $S$ to sample the answer token from that distribution. The discriminator $D$ in it's standard form takes $\boldsymbol{e}_t$, ground-truth answer $\boldsymbol{a}_t^{gt}$ and $N-1$ "negative" answers $\{\boldsymbol{a}_{t,i}^-\}_{i=1}^{N-1}$ as input, and learns an embedding space such that $\text{similarity}(\boldsymbol{e}_t, f(\boldsymbol{a}_t^{gt})) > \text{similarity}(\boldsymbol{e}_t, f(\boldsymbol{a}_{t,\cdot}^-))$, where $f(\cdot)$ is the embedding function. When we enable the communication between $D$ and $G$, we feed the sampled answer $\hat{\boldsymbol{a}}_t$ into discriminator, and optimize the generator $G$ to produce samples that get higher scores in $D$'s metric space.

We now describe each component of our approach in detail.

## 4.1 History-Conditioned Image Attentive Encoder (HCIAE)

An important characteristic in dialogs is the use of co-reference to avoid repeating entities that can be contextually resolved. In fact, in the VisDial dataset [7] nearly all (98%) dialogs involve at least one pronoun. This means that for a model to correctly answer a question, it would require a reliable mechanism for co-reference resolution.

A common approach is to use an encoder architecture with an attention mechanism that implicitly performs co-reference resolution by identifying the portion of the dialog history that can help in answering the current question [7, 38, 39, 32]. while using a holistic representation for the image. Intuitively, one would also expect that the answer is also localized to regions in the image, and be consistent with the attended history.

With this motivation, we propose a novel encoder architecture (called HCIAE) shown in Fig. 2. Our encoder first uses the current question to attend to the exchanges in the history, and then use the question and attended history to attend to the image, so as to obtain the final encoding.

Specifically, we use the spatial image features $\boldsymbol{V} \in \mathcal{R}^{d \times k}$ from a convolution layer of a CNN. $\boldsymbol{q}_t$ is encoded with an LSTM to get a vector $\boldsymbol{m}_t^q \in \mathcal{R}^d$. Simultaneously, each previous round of history $(H_0, \ldots, H_{t-1})$ is encoded separately with another LSTM as $\boldsymbol{M}_t^h \in \mathcal{R}^{d \times t}$. Conditioned on the question embedding, the model attends to the history. The attended representation of the history and the question embedding are concatenated, and used as input to attend to the image:

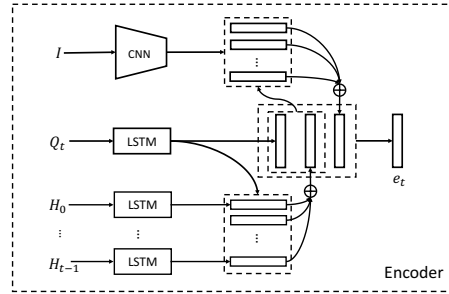

$$\boldsymbol{z}_t^h = \boldsymbol{w}_a^T \tanh(\boldsymbol{W}_h \boldsymbol{M}_t^h + (\boldsymbol{W}_q \boldsymbol{m}_t^q) \mathbb{1}^T) \quad (1)$$

$$\boldsymbol{\alpha}_t^h = \text{softmax}(\boldsymbol{z}_t^h) \quad (2)$$

**Figure 2:** Structure of the proposed encoder.

where $\mathbb{1} \in \mathcal{R}^t$ is a vector with all elements set to 1. $\boldsymbol{W}_h, \boldsymbol{W}_q \in \mathcal{R}^{t \times d}$ and $\boldsymbol{w}_a \in \mathcal{R}^k$ are parameters to be learned. $\boldsymbol{\alpha} \in \mathcal{R}^k$ is the attention weight over history. The attended history feature $\hat{\boldsymbol{m}}_t^h$ is a convex combination of columns of $\boldsymbol{M}_t$, weighted appropriately by the elements of $\boldsymbol{\alpha}_t^h$. We further concatenate $\boldsymbol{m}_t^q$ and $\hat{\boldsymbol{m}}_t^h$ as the query vector and get the attended image feature $\hat{\boldsymbol{v}}_t$ in the similar manner. Subsequently, all three components are used to obtain the final embedding $\boldsymbol{e}_t$:

$$\boldsymbol{e}_t = \tanh(\boldsymbol{W}_e [\boldsymbol{m}_t^q, \hat{\boldsymbol{m}}_t^h, \hat{\boldsymbol{v}}_t]) \quad (3)$$

where $\boldsymbol{W}_e \in \mathcal{R}^{d \times 3d}$ is weight parameters and $[\cdot]$ is the concatenation operation.

## 4.2 Discriminator Loss

Discriminative visual dialog models produce a distribution over the candidate answer list $\mathcal{A}_t$ and maximize the log-likelihood of the correct option $\boldsymbol{a}_t^{gt}$. The loss function for $D$ needs to be conducive for knowledge transfer. In particular, it needs to encourage perceptually meaningful similarities. Therefore, we use a metric-learning multi-class N-pair loss [43] defined as:

$$\mathcal{L}_D = \mathcal{L}_{n-pair}\Big(\{\boldsymbol{e}_t, \boldsymbol{a}_t^{gt}, \{\boldsymbol{a}_{t,i}^-\}_{i=1}^{N-1}\}, f\Big) = \log\left(1 + \overbrace{\sum_{i=1}^{N} \exp\Big(\underbrace{\boldsymbol{e}_t^\top f(\boldsymbol{a}_{t,i}^-) - \boldsymbol{e}_t^\top f(\boldsymbol{a}_t^{gt})}_{\text{score margin}}\Big)}^{\text{logistic loss}}\right) \quad (4)$$

where $f$ is an attention based LSTM encoder for the answer. This attention can help the discriminator better deal with paraphrases across answers. The attention weight is learnt through a 1-layer MLP over LSTM output at each time step. The N-pair loss objective encourages learning a space in which the ground truth answer is scored higher than other options, and at the same time, encourages options similar to ground truth answers to score better than dissimilar ones. This means that, unlike the multiclass logistic loss, the options that are correct but different from the correct option may not be overly penalized, and thus can be useful in providing a reliable signal to the generator. See Fig. 1 for an example. Follwing [43], we regularize the L2 norm of the embedding vectors to be small.

## 4.3 Discriminant Perceptual Loss and Knowledge Transfer from $D$ to $G$

At a high-level, our approach for transferring knowledge from $D$ to $G$ is as follows: $G$ repeatedly queries $D$ with answers $\hat{\boldsymbol{a}}_t$ that it generates for an input embedding $\boldsymbol{e}_t$ to get feedback and update itself. In each such update, $G$'s goal is to update its parameters to try and have $\hat{\boldsymbol{a}}_t$ score higher than the correct answer, $\boldsymbol{a}_t^{gt}$, under $D$'s learned embedding and scoring function. Formally, the perceptual loss that $G$ aims to optimize is given by:

$$\mathcal{L}_G = \mathcal{L}_{1-pair}\Big(\{\boldsymbol{e}_t, \hat{\boldsymbol{a}}_t, \boldsymbol{a}_t^{gt}\}, f\Big) = \log\left(1 + \exp\Big(\boldsymbol{e}_t^\top f(\boldsymbol{a}_t^{gt}) - \boldsymbol{e}_t^\top f(\hat{\boldsymbol{a}}_t)\Big)\right) \quad (5)$$

where $f$ is the embedding function learned by the discriminator as in (4). Intuitively, updating generator parameters to minimize $\mathcal{L}_G$ can be interpreted as learning to produce an answer sequence $\hat{\boldsymbol{a}}_t$ that 'fools' the discriminator into believing that this answer should score higher than the human response $\boldsymbol{a}_t^{gt}$ under the discriminator's learned embedding $f(\cdot)$ and scoring function.

While it is straightforward to sample an answer $\hat{\boldsymbol{a}}_t$ from the generator and perform a forward pass through the discriminator, naïvely, it is not possible to backpropagate the gradients to the generator parameters since sampling discrete symbols results in zero gradients w.r.t. the generator parameters. To overcome this, we leverage the recently introduced continuous relaxation of the categorical distribution – the Gumbel-softmax distribution or the Concrete distribution [18, 30].

At an intuitive level, the Gumbel-Softmax (GS) approximation uses the so called 'Gumbel-Max trick' to reparametrize sampling from a categorical distribution and replaces argmax with softmax to obtain a continuous relaxation of the discrete random variable. Formally, let $\boldsymbol{x}$ denote a $K$-ary categorical random variable with parameters denoted by $(p_1, \ldots p_K)$, or $\boldsymbol{x} \sim Cat(\boldsymbol{p})$. Let $\big(g_i\big)_1^K$ denote $K$ IID samples from the standard Gumbel distribution, $g_i \sim F(g) = e^{-e^{-g}}$. Now, a sample from the Concrete distribution can be produced via the following transformation:

$$y_i = \frac{e^{(\log p_i + g_i)/\tau}}{\sum_{j=1}^{K} e^{(\log p_j + g_j)/\tau}} \qquad \forall i \in \{1, \ldots, K\} \quad (6)$$

where $\tau$ is a temperature parameter that control how close samples $\boldsymbol{y}$ from this Concrete distribution approximate the one-hot encoding of the categorical variable $\boldsymbol{x}$.

As illustrated in Fig. 1, we augment the LSTM in $G$ with a sequence of GS samplers. Specifically, at each position in the answer sequence, we use a GS sampler to sample an answer token from that conditional distribution. When coupled with the straight-through gradient estimator [2, 18] this enables end-to-end differentiability. Specifically, during the forward pass we discretize the GS samples into discrete samples, and in the backward pass use the continuous relaxation to compute gradients. In our experiments, we held the temperature parameter fixed at 0.5.

# 5 Experiments

**Dataset and Setup.** We evaluate our proposed approach on the VisDial dataset [7], which was collected by Das *et al*. by pairing two subjects on Amazon Mechanical Turk to chat about an image. One person was assigned the role of a 'questioner' and the other of 'answerer'. One worker (the questioner) sees only a single line of text describing an image (caption from COCO [25]); the image remains hidden to the questioner. Their task is to ask questions about this hidden image to "imagine the scene better". The second worker (the answerer) sees the image and caption and answers the questions. The two workers take turns asking and answering questions for 10 rounds. We perform experiments on VisDial v0.9 (the latest available release) containing 83k dialogs on COCO-train and 40k on COCO-val images, for a total of 1.2M dialog question-answer pairs. We split the 83k into 82k for `train`, 1k for `val`, and use the 40k as `test`, in a manner consistent with [7]. The caption is considered to be the first round in the dialog history.

**Evaluation Protocol.** Following the evaluation protocol established in [7], we use a retrieval setting to evaluate the responses at each round in the dialog. Specifically, every question in VisDial is coupled with a list of 100 candidate answer options, which the models are asked to sort for evaluation purposes. $D$ uses its score to rank these answer options, and $G$ uses the log-likelihood of these options for ranking. Models are evaluated on standard retrieval metrics – (1) mean rank, (2) recall @$k$, and (3) mean reciprocal rank (MRR) – of the human response in the returned sorted list.

**Pre-processing.** We truncate captions/questions/answers longer than 24/16/8 words respectively. We then build a vocabulary of words that occur at least 5 times in `train`, resulting in 8964 words.

**Training Details** In our experiments, all 3 LSTMs are single layer with $512d$ hidden state. We use VGG-19 [42] to get the representation of image. We first rescale the images to be $224 \times 224$ pixels, and take the output of last pooling layer ($512 \times 7 \times 7$) as image feature. We use the Adam optimizer with a base learning rate of 4e-4. We pre-train $G$ using standard MLE for 20 epochs, and $D$ with supervised training based on Eq (4) for 30 epochs. Following [43], we regularize the $L^2$ norm of the embedding vectors to be small. Subsequently, we train $G$ with $\mathcal{L}_G + \alpha \mathcal{L}_{MLE}$, which is a combination of discriminative perceptual loss and MLE loss. We set $\alpha$ to be 0.5. We found that including $\mathcal{L}_{MLE}$ (with teacher-forcing) is important for encouraging $G$ to generate grammatically correct responses.

## 5.1 Results and Analysis

**Baselines.** We compare our proposed techniques to the current state-of-art generative and discriminative models developed in [7]. Specifically, [7] introduced 3 encoding architectures – Late Fusion (**LF**), Hierarchical Recurrent Encoder (**HRE**), Memory Network (**MN**) – each trained with a generative (**-G**) and discriminative (**-D**) decoder. We compare to all 6 models.

**Our approaches.** We present a few variants of our approach to systematically study the individual contributions of our training procedure, novel encoder (HCIAE), self-attentive answer encoding (ATT), and metric-loss (NP).

- **HCIAE-G-MLE** is a generative model with our proposed encoder trained under the MLE objective. Comparing this variant to the generative baselines from [7] establishes the improvement due to our encoder (HCIAE).

- **HCIAE-G-DIS** is a generative model with our proposed encoder trained under the mixed MLE and discriminator loss (knowledge transfer). This forms our best generative model. Comparing this model to **HCIAE-G-MLE** establishes the improvement due to our discriminative training.

- **HCIAE-D-MLE** is a discriminative model with our proposed encoder, trained under the standard discriminative cross-entropy loss. The answer candidates are encoded using an LSTM (no attention). Comparing this variant to the discriminative baselines from [7] establishes the improvement due to our encoder (HCIAE) in the discriminative setting.

- **HCIAE-D-NP** is a discriminative model with our proposed encoder, trained under the n-pair discriminative loss (as described in Section 4.2). The answer candidates are encoded using an LSTM (no attention). Comparing this variant to **HCIAE-D-MLE** establishes the improvement due to the n-pair loss.

- **HCIAE-D-NP-ATT** is a discriminative model with our proposed encoder, trained under the n-pair discriminative loss (as described in Section 4.2), and using the self-attentive answer encoding. Comparing this variant to **HCIAE-D-NP** establishes the improvement due to the self-attention mechanism while encoding the answers.

**Table 1:** Results (generative) on VisDial dataset. "MRR" is mean reciprocal rank and "Mean" is mean rank.

| Model | MRR | R@1 | R@5 | R@10 | Mean |
|---|---|---|---|---|---|
| LF-G [7] | 0.5199 | 41.83 | 61.78 | 67.59 | 17.07 |
| HREA-G [7] | 0.5242 | 42.28 | 62.33 | 68.17 | 16.79 |
| MN-G [7] | 0.5259 | 42.29 | 62.85 | 68.88 | 17.06 |
| HCIAE-G-MLE | 0.5386 | 44.06 | 63.55 | 69.24 | 16.01 |
| HCIAE-G-DIS | **0.5467** | **44.35** | **65.28** | **71.55** | **14.23** |

**Table 2:** Results (discriminative) on VisDial dataset.

| Model | MRR | R@1 | R@5 | R@10 | Mean |
|---|---|---|---|---|---|
| LF-D [7] | 0.5807 | 43.82 | 74.68 | 84.07 | 5.78 |
| HREA-D [7] | 0.5868 | 44.82 | 74.81 | 84.36 | 5.66 |
| MN-D [7] | 0.5965 | 45.55 | 76.22 | 85.37 | 5.46 |
| HCIAE-D-MLE | 0.6140 | 47.73 | 77.50 | 86.35 | 5.15 |
| HCIAE-D-NP | 0.6182 | 47.98 | 78.35 | 87.16 | 4.92 |
| HCIAE-D-NP-ATT | **0.6222** | **48.48** | **78.75** | **87.59** | **4.81** |

**Results.** Tables 1, 2 present results for all our models and baselines in generative and discriminative settings. The key observations are:

1. **Main Results for HCIAE-G-DIS:** Our final generative model with all 'bells and whistles', **HCIAE-G-DIS**, uniformly performs the best under all the metrics, outperforming the previous state-of-art model **MN-G** by 2.43% on R@5. This shows the importance of the knowledge transfer from the discriminator and the benefit from our encoder architecture.

2. **Knowledge transfer *vs*. encoder for $G$:** To understand the relative importance of the proposed history conditioned image attentive encoder (HCIAE) and the knowledge transfer, we compared the performance of **HCIAE-G-DIS** with **HCIAE-G-MLE**, which uses our proposed encoder but without any feedback from the discriminator. This comparison highlights two points: first, **HCIAE-G-MLE** improves R@5 by 0.7% over the current state-of-art method (**MN-D**) confirming the benefits of our encoder. Secondly, and importantly, its performance is lower than **HCIAE-G-DIS** by 1.7% on R@5, confirming that the modifications to encoder alone will not be sufficient to gain improvements in answer generation; knowledge transfer from $D$ greatly improves $G$.

3. **Metric loss *vs*. self-attentive answer encoding**: In the purely discriminative setting, our final discriminative model (**HCIAE-D-NP-ATT**) also beats the performance of the corresponding state-of-art models [7] by 2.53% on R@5. The n-pair loss used in the discriminator is not only helpful for knowledge transfer but it also improves the performance of the discriminator by 0.85% on R@5 (compare **HCIAE-D-NP** to **HCIAE-D-MLE**). The improvements obtained by using the answer attention mechanism leads to an additional, albeit small, gains of 0.4% on R@5 to the discriminator performance (compare **HCIAE-D-NP** to **HCIAE-D-NP-ATT**).

## 5.2 Does updating discriminator help?

Recall that our model training happens as follows: we independently train the generative model **HCIAE-G-MLE** and the discriminative model **HCIAE-D-NP-ATT**. With **HCIAE-G-MLE** as the initialization, the generative model is updated based on the feedback from **HCIAE-D-NP-ATT** and this results in our final **HCIAE-G-DIS**.

We performed two further experiments to answer the following questions:

- What happens if we continue training **HCIAE-D-NP-ATT** in an adversarial setting? In particular, we continue training by maximizing the score of the ground truth answer $a_t^{gt}$ and minimizing the score of the generated answer $\hat{a}_t$, effectively setting up an adversarial training regime $\mathcal{L}_D = -\mathcal{L}_G$. The resulting discriminator **HCIAE-GAN1** has significant drop in performance, as can be seen in Table. 4 (32.97% R@5). This is perhaps expected because **HCIAE-GAN1** updates its parameters based on only two answers, the ground truth and the generated sample (which is likely to be similar to ground truth). This wrecks the structure that **HCIAE-D-NP-ATT** had previously learned by leveraging additional incorrect options.

- What happens if we continue structure-preserving training of **HCIAE-D-NP-ATT**? In addition to providing **HCIAE-D-NP-ATT** samples from $G$ as fake answers, we also include incorrect options as negative answers so that the structure learned by the discriminator is preserved. **HCIAE-D-NP-ATT** continues to train under loss $\mathcal{L}_D$. In this case (**HCIAE-GAN2** in Table. 4), we find that there is a small improvement in the performance of $G$. The additional computational overhead to training the discriminator supersedes the performance improvement. Also note that **HCIAE-D-NP-ATT** itself gets worse at the dialog task.

**Table 3:** Qualitative comparison. "Ours" are samples from G-DIS model with different gumbel noise $z$. Images from the COCO dataset [25].

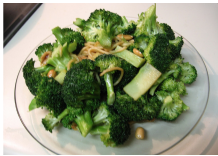   

**Q**: Is it a home or restaurant?
**A**: I think restaurant.
**G-MLE**: I can't tell.
**Ours** ($z1$): Hard to say.
**Ours** ($z2$): It looks like a restaurant.
**Ours** ($z3$): I can't tell because it is too close.

**Q**: Can you see his face?
**A**: I am not sure.
**G-MLE**: Yes.
**Ours** ($z1$): I can only see the back of his body.
**Ours** ($z2$): No.
**Ours** ($z3$): No , he's too far away.

**Q**: How old does the man seem to be?
**A**: 20's.
**G-MLE**: Late teens.
**Ours** ($z1$): On his 20s.
**Ours** ($z2$): In his twenties.
**Ours** ($z3$): Mid 20's.

**Q**: Can you see broccoli?
**A**: Yes, 3 larger pieces and some small.
**G-MLE**: No, just the broccoli.
**Ours** ($z1$): I can see broccoli, slightly butter.
**Ours** ($z2$): Yes, there is broccoli.
**Ours** ($z3$): Yes, broccoli is green.

**Table 4:** Adversarial training results on VisDial dataset.

| Model | Discriminative | | | | | Generative | | | | |
|---|---|---|---|---|---|---|---|---|---|---|
| | MRR | R@1 | R@5 | R@10 | Mean | MRR | R@1 | R@5 | R@10 | Mean |
| HCIAE-D-NP-ATT | 0.6222 | 48.48 | 78.75 | 87.59 | 4.81 | - | - | - | - | - |
| HCIAE-G-DIS | - | - | - | - | - | 0.5467 | 44.35 | 65.28 | 71.55 | 14.23 |
| HCIAE-GAN1 | 0.2177 | 8.82 | 32.97 | 52.14 | 18.53 | 0.5298 | 43.12 | 62.74 | 68.58 | 16.25 |
| HCIAE-GAN2 | 0.6050 | 46.20 | 77.92 | 87.20 | 4.97 | 0.5459 | 44.33 | 65.05 | 71.40 | 14.34 |

One might wonder, why not train a GAN for visual dialog? Formulating the task in a GAN setting would involve $G$ and $D$ training in tandem with $D$ providing feedback as to whether a response that $G$ generates is real or fake. We found this to be a particularly unstable setting, for two main reasons: First, consider the case when the ground truth answer and the generated answers are the same. This happens for answers that are typically short or 'cryptic' (e.g. *'yes'*). In this case, $D$ can not train itself or provide feedback, as the answer is labeled both positive and negative. Second, in cases where the ground truth answer is descriptive but the generator provides a short answer, $D$ can quickly become powerful enough to discard generated samples as fake. In this case, $D$ is not able to provide any information to $G$ to get better at the task. Our experience suggests that the discriminator, if one were to consider a 'GANs for visual dialog' setting, can not merely be focused on differentiating fake from real. It needs to be able to score similarity between the ground truth and other answers. Such a scoring mechanism provides a more reliable feedback to $G$. In fact, as we show in the previous two results, a pre-trained $D$ that captures this structure is the key ingredient in sharing knowledge with $G$. The adversarial training of $D$ is not central.

### 5.3 Qualitative Comparison

In Table 3 we present a couple of qualitative examples that compares the responses generated by G-MLE and G-DIS. G-MLE predominantly produces 'safe' and less informative answers, such as *'Yes'* and or *'I can't tell'*. In contrast, our proposed model G-DIS does so less frequently, and often generates more diverse yet informative responses.

## 6 Conclusion

Generative models for (visual) dialog are typically trained with an MLE objective. As a result, they tend to latch on to safe and generic responses. Discriminative (or retrieval) models on the other hand have been shown to significantly outperform their generative counterparts. However, discriminative models can not be deployed as dialog agents with a real user where canned candidate responses are not available. In this work, we propose transferring knowledge from a powerful discriminative visual dialog model to a generative model. We leverage the Gumbel-Softmax (GS) approximation to the discrete distribution –specifically, a RNN augmented with a sequence of GS samplers, coupled with a ST gradient estimator for end-to-end differentiability. We also propose a novel visual dialog encoder that reasons about image-attention informed by the history of the dialog; and employ a metric learning loss along with a self-attentive answer encoding to enable the discriminator to learn meaningful structure in dialog responses. The result is a generative visual dialog model that significantly outperforms state-of-the-art.

## Footnotes

*Work was done while at Facebook AI Research.

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
