[Reviews · NeurIPS 2017]

Reviewer 1



This paper describes an improved training procedure for visual dialogue models. Rather than maximizing the likelihood of a collection of training captions, this approach first trains a discriminator model to rank captions in a given context by embedding them in a common space, then uses scores from this discriminator as an extra component in a loss function for a generative sequence prediction model. This improved trainin procedure produces modest improvements on an established visual dialogue benchmark over both previous generative approaches as well as adversarial training. I think this is a pretty good paper, though there are a few places in which the presentation could be improved. SPECIFIC COMMENTS The introduction claims that the discriminator "has access to more information than the generator". I'm not clear on what this means. It is true that in a single update step, the discriminator here is trained to distinguish the ground truth answer from a collection of answers, while in the standard conditional GAN setting the discriminator compares only one sample at a time to the ground truth. But over the course of training a normal GAN discriminator will also get access to lots of samples from the training distribution. More generally, there's an equivalence between the problem of learning to assign the right probability to a sample from a distribution and learning to classify whether a sample came from the target distribution or a noise distribution (see e.g. the NCE literature). Based on this section, I expected the discriminator to be able to learn features on interactions between multiple candidates, and was confused when it wound up assigning an embedding to each candidate independently. Related work: given the offered interpretation of D as a "perceptual loss", I was surprised not to see any mention in the related work section of similar loss functions used image generation tasks (e.g. Johnson et al). I was not able to find anything in a quick lit search of similar losses being used for natural language processing tasks, which is one of the things that's exciting about this paper, but you should still discuss similar work in other areas. @137 it's standard form The explanation offered for the objective function at 177 is a little strange. This is a logistic regression loss which encourages the model to assign highest probability to the ground-truth caption, and I think it will be most understandable to readers if presented in that familiar form. In particular, it does not have an interpretation as a metric-learning procedure unless the norms of the learned representations are fixed (which doesn't seem to be happening here), and is not a margin loss. I don't understand what "assigned on-the-fly" @186 means. Just that you're reusing negative examples across multiple examples in the same batch? @284 modifications to encoder alone Table captions should explain that MRR -> "mean reciprocal rank" and especially "mean" -> "mean rank".

Reviewer 2



This paper proposes a novel neural sequence model for visual dialog generation. Motivated by the flexibility of generative dialog model (G) and the good performance of discriminative dialog model (D), authors propose an interesting way to combine advantages of both models by building communication between two models. This is an interesting idea. Different from the adversarial training used in GAN models, they pretrain G with MLE loss and D with a metric learning loss respectively in advance and then further train G with both metric learning loss and MLE loss. By using metric learning loss, it is encouraged to learn a semantic embedding space. The use of Gumbel-Softmax sampler makes it possible to have end-to-end training for metric learning loss. Besides, a more advanced attention model is proposed to effectively model image, history and question into an embedding vector. In the experiment section, authors have done extensive ablation study to validate the effectiveness of different components of the proposed method. There is a mistake on line 304 where it says '45.78% R@5' but I can not find this number in Table. 4. I suggest authors to mention how they choose other options for N-pair loss. Are all other 99 options are used or only a subset are sampled and how? Overall, it is a good paper which proposes a novel and effective method for the task of visual dialog generation.

Reviewer 3



This paper proposes a training procedure for visual dialog models. The paper first notes that the strength of discriminatively trained models has supervisor performance over generative models, but have the limitation of not being useful in practical settings, where novel responses are needed and no ranked list is available; and the strength of generative training is their ability to deal with unknown dialogs but have the limitation of generating safe responses, sticking to known responses. The authors seek to draw from the strengths of both of these training paradigms through knowledge transfer from a discriminative model to a generative model. For an end-to-end differentiable loss, the paper makes use of Gumbel-Softmax. A new encoder is proposed, which does not treat the image as a single entity, instead the proposed image attention model considers the fact that the question is likely to be relevant to a specific part of the image. The proposed encoder uses the question to attend to the dialog history, and then uses the question and attended dialog history to perform attention on the image. Experiments were carried out on the VisDial dataset, a visual dialog dataset generated using mechanical turk users. 1. When updating the discriminator during training of the generator, i,e. adversarial training the reason provided for bad performance is that the the discriminator in this setting updates its parameters based on the ground truth and the generated sample but does not make use of the additional incorrect options. One would imagine that the objective function of the adversarial discriminator could be augmented to also take into account the additional incorrect options, in addition to the adversarial part of the loss? 2. In terms of empirical results, although modest gains in performance are observed at R@5 and R@10 (~ 2%) it is unclear if the improvements at R@1 are significant. 3. Overall, taken individually, the contributions of the paper are mostly based on prior work, however, their combination is conceptually interesting. On the other hand, performance improvements at R@1 are not significant.